# Relationship between Health Inequalities and Breast Cancer Survival in Mexican Women

**DOI:** 10.3390/ijerph20075329

**Published:** 2023-03-30

**Authors:** Isabel Sollozo-Dupont, Victor Jesús Lara-Ameca, Dulce Cruz-Castillo, Yolanda Villaseñor-Navarro

**Affiliations:** Department of Radiology, Instituto Nacional de Cancerología México, Mexico City 14080, Mexico

**Keywords:** health iniquities, breast cancer, Mexico

## Abstract

Objective: We aimed to analyze the relationship between the survival of patients with breast cancer and health inequalities. Methods: A retrospective cohort study of women with stage III breast cancer according to public healthcare was conducted. Groups were stratified according to the course of treatment and the presence of chronic disease other than cancer. Survival functions were estimated by using the Kaplan–Meier estimator, while the Cox proportional hazards model was employed for prognostic assessment. Results: The study was performed on 964 breast cancer patients. One hundred and seventy-six patients (18.23%) died during the follow-up period and 788 (81.77%) were alive at the end of the follow-up period. Education, marital status, personal history of prior biopsies, and socioeconomic status (SES) were found to be linked with survival. However, only SES exceeded the baseline risk of mortality when the treatment cycle was interrupted (full treatment: unadjusted 4.683, *p* = 0.001; adjusted 4.888 *p* = 0.001, partial treatment: unadjusted 1.973, *p* = 0.012; adjusted 4.185, *p* = 0.001). The same effect was observed when stratifying by the presence of chronic disease other than cancer (with chronic disease adjusted HR = 4.948, *p* = 0.001; unadjusted HR = 3.303, *p* = 0.001; without chronic disease adjusted HR = 4.850, *p* = 0.001; unadjusted HR = 5.121, *p* = 0.001). Conclusion: Since lower SES was linked with a worse prognosis, strategies to promote preventive medicine, particularly breast cancer screening programs and prompt diagnosis, are needed.

## 1. Introduction

With a high incidence and mortality rate, breast cancer is the leading cause of cancer-related deaths among women in the world [1]. In Mexico, the breast cancer death rate climbed from 5.22 per 100,000 in 1970 to 9.65 per 100,000 in 2021 [2,3]. It is well known that systematic mammography screening for breast cancer is essential for early detection and treatment, which reduce mortality [4]. However, the opportunistic nature of breast cancer and lack of systematic and reliable reporting of mammography screening activity have significantly limited screening and treatment efficacy in low- and middle-income countries (LMICs) such as Mexico [5,6].

In addition, there are substantial differences in the incidence and mortality of cancer in LMICs, with socially disadvantaged cancer patients having a poorer quality of life and shorter survival rates than those who are well off [7].

The World Health Organization (WHO) defines health inequalities as systematic gaps in the health status of different population groups that have major negative social and economic consequences for both people and societies [8].

Numerous authors report a variety of inequalities in Mexico, including socioeconomic position (education, income and occupation) [9], physical environment (housing) [10], health care (access, price, and quality) [11], and ethnicity [12].

In recognition of these injustices in our society, this document provides a summary of the findings of a study that attempted to establish a link between some of these health inequalities and breast cancer mortality in order to confirm and/or provide evidence for the development of public policies that consider social factors when determining the health of a vulnerable population.

In the present study, we carefully examined the disparity in baseline health at the time of diagnosis as well as during treatment, as both factors are primarily associated with survival disparities in breast cancer, and numerous studies suggest that they are a source of confusion [13,14].

## 2. Materials and Methods

This retrospective cohort study was conducted on females diagnosed and treated at the National Institute of Cancer in Mexico between 2011 and 2016. It was approved by the Research and Ethics Committees of INCan (registration numbers 017/017/RTI and CEI/1156/17, respectively). The sample size was calculated using the sample size calculator for survival studies given at http://sample-size.net/sample-size-survival-analysis/, accessed on 1 September 2022). Both type 1 errors (alpha) and type 2 errors (beta) were estimated to be 1%, while 50% of patients were exposed. Following a previous study by Monfarred et al., 2017, we only included patients with a definitive diagnosis of stage III breast cancer under treatment with neoadjuvant chemotherapy (NAC) by using Epirubicin and/or Docetaxel [15]. Excluded were patients with a second cancer, breast cancer on the opposite side, or a serious systemic condition [15,16]. Researchers designed a questionnaire for patient interviews. After medical consultations, two fieldwork-trained nurses conducted face-to-face interviews with patients to establish if they met the inclusion criteria, if they consented to participate, and asked them to complete informed consent forms. In the medical records, two field coordinators confirmed the diagnosis and treatment of patients. Age at diagnosis, marital status, SES, level of education, health insurance coverage, breast cancer screening program adherence, place of childhood residence, health records, and breast cancer family history were the health inequalities examined in the present work. In order to control the baseline risk of mortality, we stratified all of our analysis by considering two additional variables: treatment course (full or partial) and the presence of a non-cancer chronic condition (yes or no). The specific comorbidities assessed were diabetes mellitus, hypertension, coronary heart disease (CHD), stroke, and chronic gastritis.

Financial status was assessed by using the per capita housing area (people per home/number of rooms per household, excluding bathroom and kitchen), which is a reliable and relevant metric for measuring the SES in LMICs [16,17]. It is essential to note that the recommended maximum number of people per room is two. Therefore, we used a minimum of three people per room to designate a category with a low SES [18]. Lastly, we used a questionnaire developed by the researchers to assess breast cancer screening programs compliance. This questionnaire covers the following areas: (1) awareness of the preventative significance of mammography and (2) the competition of routine mammograms [19]. See Appendix A for additional information. The benchmark for adequate breast cancer screening programmes adherence was set at 3 points.

### Statistical Analysis

Examining patients’ medical records revealed the tumor stage and the date of death.

Variables used in this research were classified as follows: age at diagnosis (≥50, <50), marital status (single, married), financial status (good-regular, bad), level of education (high school or less, academic education), health insurance coverage (yes, no), and breast cancer screening program adherence (≥3 points, <3 points). Consideration was also given to childhood residence (rural or urban), previous biopsies (yes or no), and family history of breast cancer (yes, no).

The Kaplan–Meier (KM) and Cox proportional Hazard (PH) models were used to assess the influence of putative risk variables on the survival time of breast cancer patients. Survival time was defined as the duration from the time of disease diagnosis until death or the end of the fifth year. A binary variable was used to indicate whether a cancer patient was censored or died of the disease. Input variables were entered separately into the KM model, and the log-rank test was computed for each variable. Moreover, using age-adjusted multivariable proportional hazards models, independent prognostic factors were evaluated.

The results were given as hazard ratios (HRs) accompanied by 95% confidence intervals (CIs). By stratifying all analyses in this study by comorbidities (yes/no) and course of therapy (complete/incomplete), the relative effects of patients’ baseline health on breast cancer survival disparities were controlled.

Version 26 of SPSS was used for data analysis (Statistical Package for the Social Sciences).

## 3. Results

This research was conducted on 964 breast cancer patients ranging in age from 36 to 87 years, with a mean age of 52.25 (SD = 13.0) and a median age of 54. After five years, 81.77% of breast cancer patients were still alive, while 18.23% had passed away. As stated before, all factors included in the current study were incorporated into stratified KM models separately by both the course of treatment and a non-cancer chronic condition existent prior to cancer diagnosis. Table 1 displays the analyses according to the treatment course. The cause-specific survival rate at five years was 63% in the high school or less level of education group, compared to 85% in women with an academic education. Similarly, the proportion of survivors was lower among single women and low-income groups. Unexpectedly, the probability of survival was also lower in women who had never undergone a breast biopsy before being diagnosed (previous biopsies: yes, 89.5% vs. no previous biopsies, 68.9%). Table 2 displays the results of a KM analysis stratified by the presence of comorbidities. It should be highlighted that survival probabilities were comparable to rates from the first model (KM model stratified by the course of treatment), demonstrating a survival advantage of high education and socioeconomic status, as well as being married or having received a pre-diagnosis breast biopsy.

In the second part of the analysis, the unadjusted and adjusted breast cancer mortality risk was calculated. As previously mentioned, due to the substantially non-proportional hazards, these models were stratified both by the course of treatment and the existence of comorbidities.

As can be seen in Table 3, the degree of education was a predictive factor for overall survival in women who completed treatment. Using academic education as a reference point, HR was 1.100 (95%CI 1.09–1.52; *p* = 0.041) and 1.250 (95%CI 1.031–1.519; *p* = 0.02) for the unadjusted model and the adjusted model, respectively. Furthermore, in the unadjusted model, the risk of death for single women was 1.618 (95%CI: 1.074–2.492; *p* = 0.021) more than the risk of death for married patients; however, no risk of death was seen after adjusting for age.

In contrast, we discovered that neither greater levels of education nor being a married woman always result in patients living longer when therapy was deficient.

In addition, the HR associated with poor SES among patients who completed therapy was 4.683 (95%CI = 3.221–6.167, *p* = 0.001), which was comparable to the subgroup-adjusted HR of 4.888 (95%CI = 3.325–6.364, *p* = 0.001). Similar HRs were observed for patients with incomplete treatment (unadjusted HR = 1.973, 95%CI = 1.600–1.978, *p* = 0.012; adjusted HR = 4.185, 95%CI = 1.704–10.279, *p* = 0.002).

The financial status as a quantitative variable was also incorporated in the present analysis, leading to the next results: for patients who completed therapy, the HR was 1.187 (1.029–1.370, *p* = 0.019). Meanwhile, in the age-adjusted model, the HR was 3.314 (1.325–1.452, *p* = 0.002). An increased risk of early death was found in the group of women with partial treatment: (unadjusted HR = 2.325, 95%CI = 1.787–3.415, *p* = 0.012; adjusted HR = 1.225, 95%CI = 1.019–6.540, *p* = 0.020).

Finally, women who reported never having a breast biopsy prior to diagnosis had a significantly increased risk of breast cancer-related death after the completion of therapy (unadjusted HR = 2.885, 95%CI = 1.841–3.913; *p* = 0.001; adjusted HR = 2.455, 95%CI = 1.563–4.359; *p* = 0.001). However, the self-reported history of benign breast biopsies was unrelated to survival in patients who did not complete treatment (Table 3).

Results considering the presence of chronic diseases other than cancer are demonstrated in Table 4. It can be seen that the risk of death associated with low socioeconomic income was greater in patients with and without chronic disease (with chronic disease unadjusted HR = 4.948, 95%CI = 2.459–9.981, *p* = 0.001; adjusted HR = 3.303, 95%CI = 3.131–5.699, *p* = 0.001; without chronic disease unadjusted HR = 4.850, 95%CI = 3.358–7.005, *p* = 0.001; adjusted HR = 5.121, 95%CI = 3.383–7.750, *p* = 0.001). The financial income articulated as a quantitative variable also exhibited significant results as follows: patients with chronic disease unadjusted model HR = 3.250, 95%CI = 2.180–4.520, *p* = 0.001; and adjusted model HR = 2.850, 95%CI = 1.250–3.330, *p* = 0.002. In a parallel manner, the results obtained in patients with chronic diseases were unadjusted HR = 3.328, 95%CI = 1.900–3.900, *p* = 0.02; and adjusted HR = 2.789, 95%CI = 1.850–6.505, *p =* 0.02.

There were no observed variations in the remaining variables (Table 4).

## 4. Discussion

This study revealed a declining trend in breast cancer mortality among highly educated Mexican women. Breast cancer has been documented as one of the diseases with a positive social gradient in health if a woman has high levels of education [20]. Thus, a consistent positive link between education and breast cancer mortality has been previously reported in most Western economies [21,22,23]. Multiple possible explanations exist for the observed results. Professionals and people with a high level of education tend to reside in more developed cities, where they have a greater likelihood of accessing high-quality healthcare services [24]. Additionally, an extensive list of risk factors which impact survival has been established for breast cancer. Many of these are associated with education level, such as knowledge about cancer risk, attitudes toward and access to routine mammography screening, diagnosis at earlier stages, the timeliness of treatment initiation, the receipt of appropriate treatments, and exposure to lifestyle and/or environmental factors that impact tumor progression [25,26].

However, we found that high education did not compensate for the risk of death in women who decided to decline treatment, justifying that the stratification analysis by unequal treatments must be used to validate health inequalities in breast cancer. To the best of our knowledge, a reasonable estimate of the incidence of patients refusing standard breast cancer treatments is not available at our institution. Our data exhibited that 1.5% refused standard treatments, showing poor outcomes (66.7% vs. 84.4% at 5 years). In a previous study, Verkooijen et al., 2005, reported that 0.7% of breast cancer patients registered at the Geneva cancer registry had declined any of the standard treatments offered. For these women, the disease-specific survival was significantly lower when compared to those who received standard treatment (36% vs. 75% at 10 years) [27]. Similar results were demonstrated by Joseph et al., 2012, with 1.2% of breast cancer patients refusing standard treatments and having poor outcomes (43.2% vs. 81.9%) [28]. Interestingly, in our study, the rate of patients who survived in the group of women with “incomplete/partial treatment” was higher than the ratios reported in studies for similar groups of patients. This was anticipated, given that “incomplete/partial treatment” and “refused standard treatment” are not synonymous. In this context, it is essential to mention that INCan (a medical institution of the highest standards) frequently receives many patients from rural areas of Mexico who have been referred by other hospitals when they have exhausted their treatment options. Thus, it is possible for our patients to have successfully completed their treatment plans, albeit in different facilities, as previously reported in other LMICs by Niu et al., 2019 [29]. This observation must be supplemented by additional research.

Despite the promising number of survivors in the group of women who “apparently” did not complete treatment, it is indubitable that in Mexico, there are important inequalities regarding the access to healthcare related to organizational and funding obstacles in the health system [30]. In addition, there is an inherent reluctance due to cultural barriers and cancer fatalism in Mexican women that often hampers efforts to provide effective treatments [31].

Another important finding in the present work was that married individuals have a significantly longer life expectancy than single individuals. In accordance with these findings, previous research has found that married breast cancer patients have a lower mortality rate than widowed/divorced/separated individuals and those who have never been married [32,33]. For instance, Kaplan and Kronick (2006) and Jia and Lubetkin (2020) discovered an increased risk of death among unmarried individuals compared to married individuals, and within this unmarried group, never having been married was the strongest predictor of premature mortality [34,35]. According to Rock and Zettel (2005), the protective effects of marriage may be attributable to the fact that healthier individuals are more likely to be married, and marriage may contribute to better health [36]. However, as occurred with the level of education, the differences between married and unmarried individuals tend to disappear among patients with incomplete/partial treatment. This lack of difference by marital status is not surprising given that breast cancer treatment interruption is one of the strongest and independent predictors of premature death [37].

Interestingly, women with a history of biopsies prior to being diagnosed had a lower risk of death than women without a history of biopsies. According to our knowledge, no relevant evidence supports this conclusion. However, it is well-known that women with biopsied benign breast tumors have an increased risk of developing breast cancer in the future. Specifically, a woman with atypical hyperplasia have a HR of 4.4 (95%CI; 2.7–7.1) for the development of breast cancer when compared to a woman with no known breast biopsy experience and a normal mammogram [38,39]. Thus, we hypothesize that the lowest risk of death detected here among women with a history of prior biopsies may be associated with an increased psychological burden of a benign breast biopsy, suggesting that Mexican women need to ensure early care in organized quality screening programs to reduce this burden.

Again, when we analyze the risk associated with the history of biopsies, the greatest risk was observed in women who did not complete their prescribed treatment, corroborating the increased basal risk of death for women who were treated incompletely.

Without any ambiguity, our most noteworthy conclusion was that the risk of mortality was highest among women with the lowest SES, independently of the treatment’s length. This might be connected to the influence of obstacles to the prompt detection and treatment of breast cancer among Mexican women that have been previously described in vulnerable groups [40]. Other authors have documented comparable outcomes in LMICs, but also in emerging economies [7,40]. In Finland, for example, Njor et al., 2015, identified an association between a higher SES and the overdiagnosis of low-risk breast cancer, as well as a reduced risk of incurable breast cancer and breast-cancer-related death [41]. In parallel, Dreyer et al., 2018, observed that poor and near-poor women were less likely to obtain treatment than women with a higher SES [42], whereas Kumachev indicated that a higher SES was associated with higher screening and treatment frequencies as well as higher survival rates [43].

The reason that the risk associated with SES was higher than the baseline risk for patients with inadequate therapy can be explained by the frequency with which our “wealthy patients” see other physicians at private institutions to complete their cancer treatments, confirming that financial status is the greatest disparity in our population. Hence, the aim to minimize socioeconomic disparities in breast cancer in Mexico should be integrated with the Sustainable Development Goals of the United Nations, whose guiding concept is that when every person is self-sufficient, the entire world prospers [44].

Lastly, it is widely recognized that, as in other developing countries, chronic illnesses in Mexico are substantial contributors to the degree of sickness, disability, and early death in the population [44]. We opted to stratify our data by the presence of chronic conditions other than cancer and found that this variable had no moderating influence on our findings. This is a highly troubling finding, since it indicates that SES remains a major risk factor even in the absence of chronic illnesses. Thus, Mexico’s cancer prevention and treatment initiatives must place a significant emphasis on SES and on disadvantaged persons.

Our study had several limitations. First, this is a single-center study, and the number of enrolled patients and event number are small; these factors could have weakened our observations. Moreover, we did not include lifestyle as a predictor. The main reason for this was that lower SES acts as a moderator of lifestyles in Mexican women. For example, compared to women with a higher SES, women with a lower SES are more likely to engage in excessive alcohol consumption, to be obese, and to use tobacco and other addictive drugs [45]. The above was corroborated in the statistical exploration of our data. As a result, we concentrated solely on SES, as multicollinearity among the disaggregated variables would likely lead to substantial instability in our models. However, the major limitation was that we did not consider a longitudinal design in order to separate change over time within subjects and differences among subjects (cohort effects) [46]. It is important to remember that when dealing with longitudinal data, not only do the response variables change over time, but the predictors or covariates can also change over time. Thus, the treatment of time-dependent covariates in the analysis of longitudinal data allows strong statistical inferences about dynamic relationships and provides more efficient estimators than can be obtained using cross-sectional data [47,48].

It is worth mentioning our greatest strength: the inclusion of only stage III breast cancer patients, providing a crucial control for our data. According to Monfarred et al. (2017), stage III breast cancer patients are more likely to die prematurely from breast cancer due to unfavorable health inequalities than any other stage. The fact that over 99 percent of women at stages I and II survived their cancer for at least five years after diagnosis may minimize the effect of social differences among patients, while the breast cancer death rate in stage IV patients is mostly attributed to a lower health status at the onset of the disease [15,16]. The aforementioned was also corroborated by a study from England using stage-specific analysis, in which it was found that lower survival of breast cancer among women with mid-level SES, relative to the most advantaged, was entirely explained by stage at diagnosis, while stage had no mediating effect on lower survival among the most disadvantaged patients [49].

## 5. Conclusions

This study indicated that there are substantial survival discrepancies by SES among breast cancer patients, signifying that there are still care gaps among vulnerable populations in Mexico. To reduce health inequalities in cancer, several important steps are required: (i) according to our findings, breast cancer inequalities must be fully documented in relation to financial position. Understanding the general properties of different indicators, however, is essential for all those involved in the design of studies connecting different health inequalities; (ii) Greater emphasis should be placed on prevention in general, with primary prevention offering an effective mechanism to reach the greatest proportion of a given population; and (iii) since many LMICs, including Mexico, often lack effective vital registration and disease surveillance systems and collect few data on the SES of their populations, all cancer control measures should be evaluated to determine whether they are effective in reducing health inequalities in breast cancer.

## Figures and Tables

**Table 1 ijerph-20-05329-t001:** Survival analysis stratified by the course of treatment.

	Total		Full		Partial		*p* Value
	N	Survival Rate N (%)	N	Survival Rate N (%)	N	Survival Rate N (%)	
All cases	964	788 (81.8)	820	692 (84.4)	144	96 (66.7)	
1. Level of education							
Academic education	140	119 (85.0%)	118	104 (87.9%)	22	15 (68.1%)	0.001
High school or less	824	524 (63.0%)	702	446 (63.6%)	122	73 (60.0%)	
2. Health insurance							
Yes	136	102 (75.0%)	94	73 (77.7%)	42	29 (69.1%)	0.148
No	828	686 (82.9%)	726	619 (85.3%)	102	67 (65.9%)	
3. Marital status							
Single	192	142 (74.0%)	141	107 (75.9%)	51	35 (68.7%)	0.024
Married	772	646 (83.7%)	679	585 (86.2%)	93	61 (65.6%)	
4. Previous biopsies							
Yes	604	540 (89.5%)	519	470 (90.4%)	85	70 (82.4%)	0.001
No	360	248 (68.9%)	301	222 (73.8%)	59	26 (44.1%)	
5. Financial status							
Good	640	578 (90.4%)	554	507 (91.6%)	86	71 (82.6%)	0.001
Bad	324	210 (64.9%)	266	185 (69.6%)	58	25 (43.2%)	
6. Adherence to breast cancer screening programs							
≥3 points	514	431 (83.9%)	432	370 (85.7%)	82	61 (74.4%)	0.540
<3 points	450	360 (80.0%)	388	325 (83.8%)	62	35 (56.5%)	
7. Place of residence in childhood							
Rural	194	126 (65.0%)	157	126 (80.3%)	37	27 (73.0%)	0.637
Urban	770	635 (82.5%)	663	566 (85.4%)	107	69 (64.5%)	
8. Familiar history of breast cancer							
Yes	258	210 (81.4%)	207	171 (82.7%)	51	39 (76.5%)	0.900
No	706	578 (81.9%)	613	521 (85.0%)	93	57 (61.3%)	
9. Age at diagnosis							
<50	368	308 (83.7%)	308	266 (86.4%)	60	42 (70.0%)	0.114
≥50	596	480 (80.6%)	512	426 (83.3%)	84	50 (64.3%)	

**Table 2 ijerph-20-05329-t002:** Survival analysis stratified by the presence of chronic diseases other than cancer.

	Total		With Chronic Diseases		No Chronic Diseases		*p* Value
	N	Survival Rate N (%)	N	Survival Rate N (%)	N	Survival Rate N (%)	
All cases	964	637 (66.0%)	172	111 (64.5%)	792	507 (64.0%)	
1. Level of education							
Academic education	140	119 (85.0%)	26	21 (79.5%)	114	78 (86.2%)	0.001
High school or less	824	518 (62.9%)	146	90 (61.6%)	678	429 (63.2%)	
2. Health insurance							
Yes	136	102 (75.0%)	28	22 (78.6%)	108	80 (74.1%)	0.16
No	828	686 (82.9%)	144	110 (76.4%)	684	576 (84.3%)	
3. Marital status							
Unmarried	192	142 (74.0%)	50	38 (76.0%)	142	104 (73.3%)	0.003
Married	772	646 (83.7%)	122	94 (77.1%)	650	552 (85.0%)	
4. Previous biopsies							
Yes	604	540 (89.5%)	120	98 (81.7%)	484	442 (91.4%)	0.001
No	360	248 (68.9%)	52	34 (65.4%)	308	214 (69.5%)	
5. Financial status							
Good	640	578 (90.4%)	116	100 (86.3%)	524	478 (91.3%)	0.001
Bad	324	210 (35.1%)	56	32 (57.2%)	268	178 (66.5%)	
6. Adherence to breast cancer screening programs							
≥3 points	499	417 (83.6%)	152	118 (77.7%)	347	299 (86.2%)	0.14
<3 points	485	391 (80.7%)	20	14 (70.0%)	445	357 (80.3%)	
7. Place of residence in childhood							
Rural	190	147 (77.4%)	149	112 (75.2%)	41	35 (85.4%)	0.664
Urban	774	661 (82.9%)	23	20 (87.0%)	751	621 (82.7%)	
8. Familiar history of breast cancer							
Yes	258	210 (81.4%)	36	30 (83.4%)	222	180 (81.1%)	0.625
No	706	578 (81.9%)	136	102 (75.0%)	570	476 (83.6%)	
9. Age at diagnosis							
<50	368	308 (83.7%)	64	54 (84.4%)	304	254 (83.6%)	0.159
≥50	596	480 (80.6%)	108	78 (72.3%)	488	402 (82.4%)	

**Table 3 ijerph-20-05329-t003:** Multivariate cox regression model of breast cancer–specific survival, stratifying by the course of treatment.

Model 1												
	Full Treatment (Unadjusted)			Full Treatment (Adjusted by Age)			Partial Treatment (Unadjusted)			Partial Treatment (Adjusted by Age)		
	*p* Value	HR	95%CI	*p* Value	HR	95%CI	*p* Value	HR	95%CI	*p* Value	HR	95%CI
1. Level of education												
High school or less	0.041	1.100	1.090–1.520	0.02	1.250	1.031–1.519	0.271	0.656	0.310–1.390	0.583	1.475	0.368–5.917
Academic education		1			1			1			1	
2. Health insurance												
No	0.380	1.222	0.781–1.911	0.815	1.028	0.609–1.905	0.130	1.731	0.851–3.524	0371	1.680	0.540–5.233
Yes		1			1			1			1	
3. Marital status												
Single	0.021	1.618	1.074–2.492	0.203	1.432	0.870–2.163	0.333	0.730	0.387–1.379	0.194	0.540	0.213–1.368
Married		1			1			1			1	
4. Previous biopsies												
No	0.001	2.885	1.841–3.913	0.001	2.455	1.563–4.359	0.05	1.120	0.800–1.871	0.89	1.085	0.691–9.869
Yes		1			1			1			1	
5. Financial status (qualitative)												
Bad	0.001	4.683	3.221–6.167	0.001	4.888	3.325–6.364	0.012	1.973	1.600–1.978	0.002	4.185	1.704–10.279
Good-regular		1			1			1			1	
6. Quantitative financial status (people per home/number of rooms per household)												
	0.019	1.187	1.029–1.370	0.002	3.314	1.325–1.452	0.013	2.325	1.787–3.415	0.02	1.225	1.019–6.540
7. Adherence to breast cancer screening programs												
<3 points	0.415	1.203	0.771–1.877	0.527	1.318	0.824–1.777	0.720	1.093	0.671–1.782	0.867	0.912	0.309–2.692
≥3 points		1			1			1			1	
8. Place of residence in childhood												
Rural	0.879	0.001	0.001–7.376	0.881	0.001	0.001–7.639	0.634	0.605	0.076–4.785	0.907	0.001	0.001–0.108
Urban		1			1			1			1	
9. Familiar history of breast cancer												
No	0.658	0.507	0.172–1.492	0.794	0.940	0.593–1.491	0.622	0.898	0.585–1.379	0.130	1.832	0.768–4.371
Yes		1			1			1			1	
10. Age												
<50	0.202	0.778	0.529–1.144				0.422	0.780	0.425–1.431			
≥50		1						1				

1 = reference category.

**Table 4 ijerph-20-05329-t004:** Multivariate cox regression model of breast cancer–specific survival, stratifying by preexisting chronic diseases other than cancer.

Model 2												
	Preexisting Chronic Diseases (Unadjusted)			Preexisting Chronic Diseases (Adjusted by Age)			No Chronic Diseases (Unadjusted)			No Chronic Diseases (Adjusted by Age)		
	*p* Value	HR	95%CI	*p* Value	HR	95%CI	*p* Value	HR	95%CI	*p* Value	HR	95%CI
1. Level of education												
High school or less	0.746	0.863	0.355–2.101	0.424	1.625	0.495–5.338	0.05	1.342	0.235–1.498	0.05	1.307	1.194–2.487
Academic education		1			1			1			1	
2. Health insurance												
No	0.454	1.433	0.559–3.676	0.806	0.853	0.240–3.037	0.957	1.013	0.626–1.640	0.840	0.943	0.536–1.660
Yes		1			1			1			1	
3. Marital status												
Single	0.654	0.848	0.411–1.749	0.232	1.846	0.676–5.043	0.115	1.374	0.925–2.042	0.075	1.524	0.958–2.424
Married		1			1			1			1	
4. Previous biopsies												
No	0.232	1.579	0.746–3.342	0.756	0.859	0.330–2.234	0.05	1.697	0.538–5.383	0.05	1.701	0.449–5.595
Yes		1			1			1			1	
5. Financial status												
Bad	0.001	4.948	2.459– 9.981	0.001	3.303	3.131–5.699	0.001	4.850	3.358–7.005	0.001	5.121	3.383–7.750
Good-regular		1			1			1			1	
6. Quantitative financial status (people per home/number of rooms per household)												
	0.001	3.250	2.180–4.520	0.001	2.850	1.250–3.330	0.002	3.328	1.900–3.900	0.02	2.789	1.850–6.505
7. Adherence to breast cancer screening programs												
<3 points	0.307	1.608	0.647–3.995	0.570	0.701	0.206–2.389	0.364	1.197	0.812–1.766	0.904	1.027	0.668–1.579
≥3 points		1			1			1			1	
8. Place of residence in childhood												
Rural	0.201	2.237	0.651–7.680	0.173	0.343	0.073–1.1601	0.934	0.966	0.420–2.218	0.682	1.216	0.477–3.101
Urban		1			1			1			1	
9. Familiar history of breast cancer												
No	0.143	2.003	0.791–5.077	0.039	0.252	0.068–0.930	0.506	0.879	0.601–1.286	0.761	1.073	0.682–1.688
Yes		1			1			1			1	
10. Age												
<50	0.042	0.444	0.203–0.971				0.116	0.749	0.522–1.074			
≥50		1						1				

## Data Availability

Data is unavailable due to privacy and ethical restrictions.

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
