# Peer review of "Relationship between Health Inequalities and Breast Cancer Survival in Mexican Women"

_ijerph, 2023, doi:10.3390/ijerph20075329_

Round 1

Reviewer 1 Report

Thank you very much for the opportunity to review this important and well executed study. It is of great importance that more evidence about health inequity and cancer outcomes in LMICs is generated and published. The authors were very thorough in their analysis and documentation.

Below are a few suggestions to further improve the analysis and interpretation of the findings:

·      The authors use the number of people per room as a proxy variable for SES which is entirely appropriate. I would suggest ton see whether using number of people per room as a continuous variable generates a positive statistically significant estimate. This would suggest that greater levels of deprivation are associated with greater inequity and greater health risks, and justify a particular focus on the most deprived individuals.

·      I would also suggest to supplement the number of persons per room variable by neighborhood of residence, provided this information is available, as a categorical variable distinguishing wealthy from poor neighborhoods (possibly a few categories in between). The motivation is that with the current approach there could be e.g. single person living in very deprived conditions in one room which would classify them as high SES according to the current SES measure. If a second SES variable based on neighborhood SES can be generated, I would also suggest to include an interaction term between number of persons per room and neighborhood SES.

·      The authors report that “Unexpectedly, the 118 probability of survival was also lower in women who had never undergone a breast biopsy before to being diagnosed”. This is an unexpected findings. One explanation I can think of is that prior biopsies may also be a proxy for lower access to (preventative) healthcare and/or lower quality healthcare (e.g. doctors actually referring patients for biopsies), similar to the “breast cancer screening programmes adherence” variable that the authors also include.

·      I would suggest that the observe lack of significance of marriage and education for the incomplete treatment group suggests that these factors may actually be predictors of completing treatment, which then in turn affects risk of death. It could be worthwhile to investigate a model where marriage and education are predictors of treatment completion.

·      My most substantial suggestion would be to add to the authors interpretation of the findings about the effect of SES. In addition to prevention and treatment adherence, SES may also be associated with other factors that affect cancer mortality, e.g. access/quality of food, housing infrastructure (e.g. cooling/heating), opportunity to rest, etc. The authors state that they find “no mediating effects” by the chronic diseases variable. Further they state: “Simply, we confirmed that a low SES increases the risk of breast cancer death” and label this a “positive result”. I find that this is actually a very concerning result, because vice versa it means that even without chronic conditions, SES is still an important risk factor. This means that SES and with that a strong focus on disadvantaged individuals has to be a major focus of cancer prevention and treatment programs in Mexico. I would suggest that the authors add a discussion about the substantial importance of serving low-income populations to the interpretation of their findings. Similarly, this should be added to the conclusion of the abstract which currently states: “Since lower SES had the worse prognostic, strategies to promote preventive medicine, particularly breast cancer screening programs and prompt diagnosis, are needed.” The findings suggest that effective strategies have to be identified such that disadvantaged individuals can access such preventive programs. In addition, cancer prevention needs to incorporate strategies that address the negative effects of low SES beyond prevention and treatment adherence.

·      A minor remark: The following sentence misses some words at the end: “The reason that the risk associated with socioeconomic status was greater than the baseline risk for patients with inadequate therapy can be explained by the frequency with which our "”

Author Response

We appreciate you taking the time to evaluate our article and provide insightful feedback. Your informative and important feedback led to potential enhancements to the current edition. The writers have carefully reviewed the comments and made every effort to respond to each one. We trust that the revised manuscript meets your high requirements.

Thank you very much for the opportunity to review this important and well executed study. It is of great importance that more evidence about health inequity and cancer outcomes in LMICs is generated and published. The authors were very thorough in their analysis and documentation.

We are very grateful to the reviewer, as his or her comments greatly enriched our discussion. 

Below are a few suggestions to further improve the analysis and interpretation of the findings:

The authors use the number of people per room as a proxy variable for SES which is entirely appropriate. I would suggest ton see whether using number of people per room as a continuous variable generates a positive statistically significant estimate. This would suggest that greater levels of deprivation are associated with greater inequity and greater health risks, and justify a particular focus on the most deprived individuals.

This comment is appropriate, and we include the proper modification in the data analysis of the new version of the manuscript. It is important to mention that we feared that the coefficients would suffer important changes due to a collinearity phenomenon. However, this was not the case, which corroborates that financial status, in its multiple dimensions, must be considered in the Mexican population to address the issue of social inequities in cancer.  It is evident that the economic gaps and lags that our population suffers today have a negative impact on the approach to the different neoplasia.

I would also suggest to supplement the number of persons per room variable by neighborhood of residence, provided this information is available, as a categorical variable distinguishing wealthy from poor neighborhoods (possibly a few categories in between). The motivation is that with the current approach there could be e.g. single person living in very deprived conditions in one room which would classify them as high SES according to the current SES measure. If a second SES variable based on neighborhood SES can be generated, I would also suggest to include an interaction term between number of persons per room and neighborhood SES.

We cannot include this variable because, although we know the neighborhood, we would have no way of classifying it by social class. In Mexico many people are incorporated into the informal sector of the economy, which distinguishes them as having a good economic position. However, their place of residence is not good as a screen because there is no way to prove income. ·      

The authors report that “Unexpectedly, the 118 probability of survival was also lower in women who had never undergone a breast biopsy before to being diagnosed”. This is an unexpected findings. One explanation I can think of is that prior biopsies may also be a proxy for lower access to (preventative) healthcare and/or lower quality healthcare (e.g. doctors actually referring patients for biopsies), similar to the “breast cancer screening programmes adherence” variable that the authors also include.

We are very grateful for these comments; the following modification was included in the discussion: The high psychological burden of a benign breast biopsy increases significantly over time in our population due to the lack of breast cancer screening programs. To reduce this burden, women need to ensure early care in organized, quality screening programs.

I would suggest that the observe lack of significance of marriage and education for the incomplete treatment group suggests that these factors may actually be predictors of completing treatment, which then in turn affects risk of death. It could be worthwhile to investigate a model where marriage and education are predictors of treatment completion.

Undoubtably, marital status as well as marital satisfaction are predictors of the course of treatment in several types of cancers, including breast cancer. We are currently building the prediction model proposed by the reviewer without incorporating it into this version of the manuscript.

My most substantial suggestion would be to add to the authors interpretation of the findings about the effect of SES. In addition to prevention and treatment adherence, SES may also be associated with other factors that affect cancer mortality, e.g. access/quality of food, housing infrastructure (e.g. cooling/heating), opportunity to rest, etc. The authors state that they find “no mediating effects” by the chronic diseases variable. Further they state: “Simply, we confirmed that a low SES increases the risk of breast cancer death” and label this a “positive result”. I find that this is actually a very concerning result, because vice versa it means that even without chronic conditions, SES is still an important risk factor. This means that SES and with that a strong focus on disadvantaged individuals has to be a major focus of cancer prevention and treatment programs in Mexico. I would suggest that the authors add a discussion about the substantial importance of serving low-income populations to the interpretation of their findings. Similarly, this should be added to the conclusion of the abstract which currently states: “Since lower SES had the worse prognostic, strategies to promote preventive medicine, particularly breast cancer screening programs and prompt diagnosis, are needed.” The findings suggest that effective strategies have to be identified such that disadvantaged individuals can access such preventive programs. In addition, cancer prevention needs to incorporate strategies that address the negative effects of low SES beyond prevention and treatment adherence.

Everything described in this paragraph is of utmost importance and was incorporated into the discussion in the new version of the manuscript. Thank you.

A minor remark: The following sentence misses some words at the end: “The reason that the risk associated with socioeconomic status was greater than the baseline risk for patients with inadequate therapy can be explained by the frequency with which our "”

Many thanks to the reviewer. The mentioned sentence was corrected.

Reviewer 2 Report

Important subject to close the gap in cancer care inequalities. But statistical analysis are not executed in the proper way. A time related model should be used. Major discrepancy in group sizes with the risk of statistical inappropriate outcomes. The fact that lifestyle factors and tumor and treatment related prognostic factors could not be considered taken into account makes the results unreliable, regrettably.

Author Response

We appreciate you taking the time to evaluate our article and provide insightful feedback. Your informative and important feedback led to potential enhancements to the current edition. The writers have carefully reviewed the comments and made every effort to respond to each one. We trust that the revised manuscript meets your high requirements.

  1. Statistical analysis are not executed in the proper way. A time related model should be used.

We would like to thank the reviewer for this excellent observation. Nonetheless, it is important for us to recognize that, in México, income inequality and gender gaps remain high over the past three decades. Especially, women suffering discrimination. It is regrettable that our population does not experience significant achievements neither in its economy nor in education over a five-year period. In this sense, the treatment we gave to the data corresponded to a cross-sectional study, pretending that the covariates would change over time as in a longitudinal model. Formally one can think of a fixed covariate as a time-dependent covariate that happens to have a constant value for all time points (https://www.annualreviews.org/doi/pdf/10.1146/annurev.publhealth.20.1.145). But we agree with you that temporal evolution is an important dimension of our methodological processes and this should be included in the modeling evaluation. Undoubtedly, in the future, we will to explore our results by using a logistic regression model for longitudinal data with time-dependent covariates. In the new version of the manuscript we discuss this as a limitation of our study, adding the following paragraph: One major limitation of our study was that we did not consider a longitudinal design in order to separate change over time within subjects and differences among subjects (cohort effects) [Diggle P, Heagerty P, Liang K, Zeger S. Analysis of Longitudinal Data. Oxford University Press: Oxford, United Kingdom, 2002.]. It is important to remember that when dealing with longitudinal data not only do the response variables change over time, but the predictors or covariates can also change over time. Thus, the treatment of time-dependent covariates in the analysis of longitudinal data allows strong statistical inferences about dynamic relationships and provides more efficient estimators than can be obtained using cross-sectional data [Zeger SL, Liang KY. Longitudinal data analysis for discrete and continuous outcomes. Biometrics 1986; 42(1):121–130].

2.The fact that lifestyle factors and tumor and treatment related prognostic factors could not be considered taken into account makes the results unreliable, regrettably.

Low SES has been linked to an increased risk to die prematurely as a result of breast cancer. But this increased risk is not related to lower SES itself. It’s related to differences in risk factors found in women with different income levels. For example, compared to women with a higher SES, women with a lower SES are more likely to engage in excessive alcohol consumption, to be obese, and to use tobacco and other addictive drugs. Perhaps we neglected to emphasize that all these characteristics are associated with an increased risk of mortality in our patients; but, as we previously mentioned, lower SES acts as a moderator of lifestyles, which has been demonstrated in our population (https://www.oecd.org/economy/surveys/Mexico-2017-OECD-economic-survey-overview.pdf). As a result, we concentrated solely on SES, as multicollinearity among the disaggregated variables would likely lead to substantial instability in the coefficient estimates. With this strategy, we were able to frame addressing severe poverty in Mexico as a primary breast cancer prevention strategy.

Undoubtedly, lifestyles will improve as a result of overcoming extreme poverty. Yet, it must be stressed that this must take place among 80% of our population, as just 20% are neither poor nor vulnerable.

We incoporate a part of the text aforementioned in the discussion of the nuew versión of our manuscript.

3.The fact that lifestyle factors and tumor and treatment related prognostic factors could not be considered taken into account makes the results unreliable, regrettably.

Thanks for your comment. Our study just focused on evaluating factors considered as potentially contributing to socio-economic inequalities in breast cancer overall survival. This decision was made based on literature indicating that tumor characteristics and treatment did not contribute to lower survival in disadvantaged women. For example, studies from the Netherlands and Sweden  found that the observed lower survival among disadvantaged women, or those with low level of education, was not explained by variation in stage of breast cancer at diagnosis, other tumor characteristics, number of nodes examined or the treatment received [1,2]. On the other hand, you are missing a piece of important information: we conducted our study with patients in stage III, exclusively. Moreover, we ensured that the treatment of all patients included in this analysis was as homogeneous as possible. In this sense we had a significant reduction in the sample available, but we are confident that we controlled for possible biases due to clinical and treatment parameters.

The above facts are reinforced in the discussion of the new version of the manuscript.

1.Bastiaannet E, De Craen AJM, Kuppen PJK, et al. Socioeconomic differences in survival among breast cancer patients in the Netherlands not explained by tumor size. Breast Cancer Res Treat. 2011;127(3):721–727.

2.Eaker S, Halmin M, Bellocco R, et al. Social differences in breast cancer survival in relation to patient management within a National Health Care System (Sweden). Int J Cancer. 2009;124(1):180–187.

It is important to mention that the new version of the manuscript was reviewed by a specialist in text editing: Ricardo Iván Román Torres.

Round 2

Reviewer 2 Report

Without inclusion of tumor- and treatment characteristics no conclusion can be drawn from this study. 
Find example of retrospective breast cancer studies in specific Cancer journals / Lancet / BMJ etc and maybe ask those authors for help. A large set of prognostic factors exist. You must include the most important ones, otherwise your cannot draw conclusion from your study. Endpoints must me time related if time is a variable, for instance when analysing survival. I would ad treatment response, compliance to treament, tumor controle and toxicity to your endpoints. Good luck.

Author Response

Following the recent comments of one of our reviewer, there are misconceptions that need to be addressed in our paper. First, we are study health inequalities affecting survival based on SES, principally. SES has been defined as potential or realized access to resources in three major domains: material, human, and social capital. Thus, it is not surprising that a relationship between socioeconomic status and health has persisted across time, place, and clinico-pathologic characteristics of diseases (Fiscella, K., & Williams, D. R. (2004). Health disparities based on socioeconomic inequities: implications for urban health care. Academic Medicine, 79(12), 1139-1147). In the search for more advanced explanations correlating low SES with poor both health and prognosis, merge the concept of social inequality. In the cancer continuum, social inequalities refer to systematic differences between social groups that affect people’s risk of treatment toxicity, high tumor stages, etc. 

When we started our data collection and analysis, we were confronted with the fact that there are several approaches to measure wealth, so we were unable to establish social groups. Our main limitation is that in LIMC there are several measures of SES, including education and/or income and/or type of job, etc. Once a suitable measure is found, SES is usually described as low, medium, and high, which offer researches possibilities to study social disparities. 

It is important to us emphasize that the social stratification processes in many LMIC it is very difficult and it differs considerably from those in HIC due to the importance of the informal economy and the lack of welfare states in many LMIC. Take this into account we prefer, in this paper, we limit ourselves to establish a relationship of some social factors with breast cancer survival in order to find variables that allow us to discovery the most vulnerable social class in our population. 

However, the effect of the factors found here affecting survival in all stage at diagnosis according to diagnostic activity, tumor properties and treatment features, will be reviewed in the near future following the methodology of Eaker et al. Consequently, debate on how low SES can be targeted is warranted among Mexican women.

The tables presented in the support document (table I and II) are an example of the study of social inequlities, because what we understand is that the reviewer is asking us to measure systematic differences between social groups relative to treatment, tumor type and tumor stage; but our aim is to find domains that allow us to establish health inequalities as a measure of social vulnerability, which Is carified in the new versión of the manuscript.
